# Brain-Derived Neurotrophin and TrkB in Head and Neck Squamous Cell Carcinoma

**DOI:** 10.3390/ijms20020272

**Published:** 2019-01-11

**Authors:** József Dudás, Anna Riml, Raphaela Tuertscher, Christian Pritz, Teresa Bernadette Steinbichler, Volker Hans Schartinger, Susanne Sprung, Rudolf Glueckert, Anneliese Schrott-Fischer, Lejo Johnson Chacko, Herbert Riechelmann

**Affiliations:** Department of Otorhinolaryngology, Medical University of Innsbruck, Anichstrasse 35, A-6020 Innsbruck, Austria; anna.riml@gmx.net (A.R.); raphaela111@gmx.net (R.T.); christianpritz@gmail.com (C.P.); teresa.steinbichler@i-med.ac.at (T.B.S.); volker.schartinger@i-med.ac.at (V.H.S.); susanne.sprung@i-med.ac (S.S.); rudolf.glueckert@i-med.ac.at (R.G.); annelies.schrott@i-med.ac.at (A.S.-F.); lejo.johnson@i-med.ac.at (L.J.C.); herbert.riechelmann@i-med.ac.at (H.R.)

**Keywords:** epithelial–mesenchymal transition, vimentin, cytokeratin, p53 mutation, paraffin embedding, riboprobe

## Abstract

We hypothesized that in head and neck squamous cell carcinoma (HNSCC), the neurotrophin brain-derived neurotrophic factor (BDNF) and its high affinity receptor TrkB regulate tumor cell survival, invasion, and therapy resistance. We used in situ hybridization for *BDNF* and immunohistochemistry (IHC) for TrkB in 131 HNSCC samples. Brain-derived neurotrophic factor was highly expressed in normal mucosa in HNSCC tissue and in cell lines, whereas only 42.74% of HNSCC tissue was TrkB^+^. One fourth of HNSCC cases was human papilloma virus (HPV)^−^ positive, but the TrkB IHC frequency was not different in HPV-positive (HPV^+^) and negative cases. The UPCI-SCC090 cells expressed constitutive levels of TrkB. Transforming-growth-factor-β1 (1 ng/mL TGF-β1) induced TrkB in a subpopulation of SCC-25 cells. A single 10-µg/mL mitomycin C treatment in UPCI-SCC090 cells induced apoptosis and BDNF did not rescue them. The SCC-25 cells were resistant to the MMC treatment, and their growth decreased after TGF-β1 treatment, but was restored by BDNF if it followed TGF-β1. Taken together, BDNF might be ineffective in HPV^+^ HNSCC patients. In HPV^−^ HNSCC patients, tumor cells did not die after chemotherapeutic challenge and BDNF with TGF-β1 could improve tumor cell survival and contribute to worse patient prognosis.

## 1. Introduction

The neurotrophin and neurotrophin receptor signal path was initially identified in neural cells. Recently, they were detected in some cancers in association with invasiveness [1]. The uptake of neurotrophins and their biological signaling is initiated following their binding to cell surface proteins and receptor dimerization [2]. The receptors are either of low affinity (low-affinity nerve growth factor (LNGFR)/p75 neurotrophin receptor (p75NTR)) [3,4] or are of high affinity Trk (tropomyosin-related kinase) receptor tyrosine kinases. The Trk family of receptors includes the neurotrophin receptor tyrosine kinase-2 (*NTRK2*) (TrkB) specific for brain-derived neurotrophic factor (BDNF) and neurotrophin-4 (NT-4) [5]. Both BDNF and TrkB are key factors in the development and maintenance of hearing function [6,7,8,9,10]. Brain-derived neurotrophic factor and its receptor moieties are instrumental to facilitate sensory neuronal survival in embryonic rat [11].

The BDNF/TrkB axis is overexpressed in neurogenic and non-neurogenic tumors, as well, its higher expression was reported to be associated with unfavored prognosis in cancer patients. Mechanistic explanations for this issue are related to tumor growth, invasion, metastasis, epithelial-to-mesenchymal transition, and resistance to chemotherapy [12,13].

A widely discussed issue is the dual proangiogenic action of BDNF, which has been claimed by Kermani and Hempstead in 2007 [14] considering both the local activation of TrkB receptors expressed on a subpopulation of endothelial cells and the recruitment of bone marrow-derived cells. This dual activity is a major contributor to neoangiogenesis [14]. Xu et al. in 2015 elegantly demonstrated that both BDNF and TrkB are essential in the angiogenesis, and are related to the endothelial cell survival process [15]. With the major role of the BDNF/TrkB axis in angiogenesis and in its relation to endothelial cell survival, several research groups agree, it is indeed an important contributor to tumor progression. In contrast, this article addresses more the function of the BDNF/TrkB axis directly in the tumor cells of the head and neck squamous cell carcinoma (HNSCC). The involvement of BDNF and its receptor TrkB in HNSCC was first indicated by Kupferman and colleagues [16], where induction of cell migration and the epithelial–mesenchymal transition (EMT) were reported. In an earlier study, the invasive properties of TrkB-expressing HNSCC cells were not significantly altered after treatment with 10 or 100 ng/mL BDNF [17]. In our previous study we focused on the interaction between fibroblasts and squamous cell carcinoma cells of HNSCC. In a co-culture model, we evidenced that inflammatory cytokines are upregulated in tumor cells co-cultured with fibroblasts, and at the same time, the fibroblasts were activated and produced increased levels of BDNF. The direct inflammatory cytokine path in BDNF regulation was also proven in this study [18,19]. Activated fibroblasts, which also produce increased levels of BDNF, were capable of induction of EMT in tumor cells [18]. In our recent study [20], we found that the conditioned medium of cultured fibroblasts in serum-deprived conditions support (i) cell proliferation, (ii) cell migration, and (iii) EMT in tumor cells of HNSCC. This conditioned medium produces an array of factors that are responsible for different functions. The most abundant growth factor of this conditioned medium is interleukin-6, which upregulates cell proliferation (unpublished data), whereas transforming growth factor-beta1 (TGF-β1) is effective in as low as 1 ng/mL concentration, and is mainly involved in the dissemination of small cell groups from the tumor cell nest [20]. Interestingly, TGF-β1 was not supporting either cell proliferation or cell survival of SCC-25 cells and was only related with cell dissemination and downregulation of epithelial cell adhesion molecules, which is an EMT-like process. In the case of BDNF cell survival [21] and cell invasion [16], inducing functions are discussed in current references, nevertheless, the cell invasion induction of HNSCC cells was doubted by Zhu et al. [17].

The newest data from the Kupferman group [22] further confirms our previous studies and claims that conditioned media derived from patient-derived carcinoma associated fibroblasts (CAFs) promotes HNSCC cell proliferation, in vitro cell migration, cell invasion, and chemotherapy resistance. Furthermore, they claim CAF and tumor cell interaction to be responsible for lymphovascular metastasis and mechanistically demonstrate the critical importance of BDNF-TrkB signaling in this process [22].

A further issue to be introduced here is a BDNF-independent oncogenic function of TrkB kinase based on gene fusion *PAN3–NTRK2*, which is as rare as one case in 411 in HNSCC [23]. Both the role of the BDNF/TrkB axis in chemotherapy resistance and lymphovascular metastasis of HNSCC [13,16,22] and a potential of oncogenic gene fusion of neurotrophin receptors [23] warrant investigation into the rationale for *BDNF/NTRK2*-targeted therapy in the treatment of HNSCC.

The development of HNSCC is not only related to tobacco use or alcohol consumption, which are the most common risk factors, but also to human papilloma virus (HPV), a sexually transmitted infection, which has an increasing significance in this field [24]. In particular, HPV-positive tumors not only have a different etiology, but they also remarkably better respond both to radio(chemo)therapy [25], and to surgery [26], compared to HPV-negative tumors. It is now a valid conclusion that HPV-positive patients have better prognosis.

In our hypothesis, BDNF might be a factor with similar function as TGF-β1 in influencing the separation of disseminating tumor cell groups but it is extended also by cell survival support functions. We presume that in HPV-positive and negative HNSCC, the BDNF/NTRK2 axis might show different significance. This hypothesis is tested now using tissue material of over 100 HNSCC patients and by experimental approaches on HPV-positive and -negative HNSCC cell lines.

## 2. Results

### 2.1. Gene Expression of BDNF and NTRK2 and Protein Synthesis of NTRK2 in Oral Mucosa and in HNSCC Tumor Tissue

Using RNA isolation, reverse transcription, PCR, and in situ hybridization, the mRNA expression of *BDNF* and *NTRK2*, and using immunohistochemistry, the protein synthesis of the NTRK2 gene product TrkB were investigated in normal mucosa from uvulopalatopharyngoplasties (UPPP) and in HNSCC tumor tissue. Brain-derived neurotrophic factor was detected at mRNA level in normal mucosa and in HNSCC tumor tissue by RT-PCR (Figure 1E) and by in situ hybridization (Figure 1A–C). Brain-derived neurotrophic factor was produced in tumor cell nests as well as in stroma (Appendix A), as displayed in Figure 1E, it was sufficiently available either in normal or in tumor tissue and also in cultured HNSCC cell lines (follows later).

Interestingly, a more limited pattern was the availability and distribution of the specific high affinity BDNF receptor, TrkB. One-hundred-and-thirty-one HNSCC samples were available for TrkB immunostaining; 75 of 131 (57.25%) HNSCC samples showed negative reaction for TrkB and 56 of 131 (42.74%) HNSCC samples showed positive reaction. TrkB positive reaction was focal (Appendix A) in 30 of 131 HNSCC (22.90%) and diffuse within the tumor cell nest (Appendix A) in 26 of 131 HNSCC (19.84%). In control normal mucosa from UPPP, only 1 of 12 (8.33%) samples showed a focal TrkB reaction (Table 1).

In a further step it was investigated if the detection of TrkB protein was a result of a normal gene product or variant *NTRK* rearrangements as published in 2018 by Rudzinsky et al. [27]. The anti-TrkB rabbit monoclonal antibody (clone 80G2) from Cell Signaling Technologies revealed positive reaction in 42.74% of HNSCC tissue, the so-called pan-Trk antibody (EPR17341 by Abcam) suggested by Rudzinsky et al. for IHC of the protein products of *NTRK* rearrangements did not detect any positive reaction in any HNSCC tissue. The positive reaction of 80G2 was further confirmed by PCR amplification of the whole protein-coding exome region of NTRK2 and by Sanger sequencing of the PCR product. In this regard, we consider the IHC reaction of the 80G2 rabbit monoclonal antibody as reliable, whereas, the EPR17341 might be limited to detect rearranged NTRK1 gene products, as published by Rudzinsky et al. [27]. The sequences received by Sanger sequencing reads were identical with wild-type *NTRK2* and aligned with more transcript variants. These data suggest that in our HNSCC material there were no *NTRK2* sequence rearrangements.

### 2.2. HPV Carcinogenesis Effect on Patient Survival, TrkB Staining Pattern Relation to HPV Carcinogenesis, Relation of TrkB Staining with HNSCC Clinical Properties

Human-papilloma-virus-positive cases were decided by immunohistochemical staining pattern of the surrogate marker p16^INK4^ being in at least 66% of the tumor cells positive [28]. Taking HPV DNA PCR analysis as the reference method, the sensitivity of p16 IHC was 78% and the specificity was 79% [29]. The p16^INK4^—based HPV evaluation was achieved in all 131 cases. Thirty-three of 131 cases (25.2%) were HPV-positive and 98 cases (74.8%) were HPV-negative. The HPV-positive cases showed significantly better survival (*p* = 0.015 by log-rank (Mantel–Cox) pairwise comparison as displayed in the Kaplan–Mayer survival curves (Appendix A). In the HPV^−^ patient group TrkB was stained in 46 of 98 patients (46.97%), and 21 patients (21.42%) had a high positive diffuse staining pattern. In the HPV^+^ patient group, TrkB was stained in 10 of 33 patients (30.33%) and 5 patients (15.15%) had high positive diffuse staining pattern. Comparing this staining distribution in HPV^+^ and HPV^−^ negative patient groups by Chi-square test, it did not show a significant difference (*p* = 0.241). Considering negative, focal, and diffuse staining patterns, there was no significant patient survival difference if all patients or HPV^+^ or HPV^−^ patients were investigated. Comparing the frequency of TrkB–stained cases among the whole collective with clinical or molecular biological patient characteristics, the following ones showed relation with TrkB antibody immunohistochemical reaction. As mentioned above, TrkB was detected in 56 of 131 HNSCC cases, and was scored semi-quantitatively as described in the Materials and Methods section. N-stage data was available in 96 cases and TrkB frequency showed a tendency of increase in along with N1–N3 (Figure 2A). More interestingly, by comparing primary (*n* = 97) and secondary or recurrent HNSCC (*n* = 23), the TrkB staining was significantly more frequent in secondary or recurrent HNSCC than in primary tumor (Figure 2B). As it was published before, a scattered TP53 staining (using the diagnostic antibody clone Bp53-11 [28]) is related with normal (wild-type) genetic background with no p53 mutations [30], while no staining or increased (over 66% of tumor cells stained) staining pattern is related with “altered”, frequently even mutated p53 [30]. Comparing HNSCC with normal (wild-type; *n* = 54) TP53 immunohistochemistry with HNSCC with altered (*n* = 76) TP53 immunohistochemistry, the TrkB staining was significantly more frequent in cases with altered TP53 than in cases with normal TP53 (Figure 2C).

Taken together, comparison of *BDNF* growth factor and *NTRK2* receptor gene expression with clinical data revealed that BDNF is produced both in stroma and in tumor cell nests of HNSCC, the availability of the *NTRK2* gene product, TrkB, is limited to approximately 40% of the cases. TrkB is present in 47% of HPV^−^ and 30% of HPV^+^ HNSCC. The TrkB staining pattern or its presence or absence did not show relation with patient survival, but TrkB staining was visibly more frequent in cases with higher levels of lymph node metastasis, was significantly more frequent in cases with secondary tumor or recurrence, and with altered p53. These data are consistent with the published evidence that the BDNF (which is sufficiently available in HNSCC)–TrkB (the receptor is available in 40% of the cases, frequently focally localized at the border of the tumor cell nests) system might be related with cell invasivity, and improved tumor cell survival in therapeutic conditions [13,16,18,19,22]. The above-mentioned clinical material-based results justified the investigation of the BDNF–TrkB axis in relation to cell migration, cell survival, and therapy resistance in available SCC-25, Detroit 562, and UPCI-SCC090 cells.

### 2.3. Gene Expression of BDNF and NTRK2, and Protein Synthesis of TrkB in SCC-25, Detroit 562, and UPCI-SCC090 Cells

In our laboratory, SCC-25 oral, Detroit 562, and UPCI-SCC090 pharyngeal HNSCC cells were available. Both SCC-25 and Detroit 562 cells were HPV-negative and UPCI-SCC90 cells were HPV-positive [28]. The SCC-25 cells were originally isolated from the primary tumor of a patient with tongue carcinoma [18,31]. The SCC-25 cells are from primary oral SCC, maintained in in vitro cultures, might be also xenografted, but they grow only in severe combined immunodeficiency (SCID) mice and not in athymic mice. The SCC-25 cells did not show metastatic potential in mouse xenograft models [32]. In vitro, SCC-25 cells were published to be radioresistant [33]. Detroit 562 cells are metastatic HPV-negative HNSCC cells, which demonstrated radio and chemoresistance [34,35], and they are also a potential metastatic cell line in xenograft models [36]. Detroit 562 cells were originally isolated from the malignant pleural effusion of a pharyngeal squamous cell carcinoma [37,38]. The UPCI-SCC090 cell line was established by Robert Ferris and co-workers [39]. The UPCI-SCC090 cells contain genome integrated HPV-16 DNA. The UPCI-SCC090 cells synthesize both p16^INK4^ surrogate HPV marker protein and E6, E7 HPV oncogene product proteins [39]. These cell lines enabled the investigation of primary and metastatic as well as HPV-positive and negative HNSCC models.

Brain-derived neurotrophic factor gene expression was detected in both SCC-25 and UPCI-SCC090 HNSCC cell lines (Figure 3A), *NTRK2* was high expressed in UPCI-SCC090 cells (Figure 3A,B,E), and low expressed in SCC-25 cells (Figure 3A,B,D). The whole 2528 base pairs PCR product was amplified in UPCI-SCC090 cells (Figure 3B), while in SCC-25 cells only a 600 base pairs PCR product was amplified (Figure 3A,B). For the amplification of the whole protein coding region it was required that all cells express *NTRK2* (Figure 3A,B,D), while for the 600 base pairs product also a very scattered *NTRK2^+^* cell population was sufficient (Figure 3A–C). *ACTB* was used as loading control (housekeeping gene), and the densitometric values of *ACTB* were used for the normalization of both BDNF and NTRK2 optical densities. At protein level using paraffin embedded cells, TrkB was not present in Detroit 562 (Figure 3C), and was only in few scattered cells in SCC-25 (Figure 3D, arrow), but was present in all cells in UPCI-SCC090 (Figure 3E) cell line.

### 2.4. Effects of BDNF Treatments in UPCI-SCC090 Cells

Since in our availabilities only UPCI-SCC090 cells were significantly TrkB^+^ at protein level, we considered this cell line at first for further investigation. MTT assay—cell growth analysis by measuring the percentage of covered area in culture dish by cells—cell counting, and scratch assay were considered for the investigation of BDNF effects on UPCI-SCC090 cells. The range of BDNF treatment concentrations followed published references and was in the range of 0–50 ng/mL [13]. We have seen effects in the MTT-assay and by measuring the covered area by cells by 25 ng/mL concentration, which did not increase by higher concentrations (Appendix A). For analysis of cell migration, the so-called scratch assay was used [20], which contains a Mitomycin C (MMC) treatment (a single 10 µg/mL treatment for 30 min at 37 °C to ensure cell cycle arrest) [40]. It is particularly important to investigate cell migration and not cell division, since these both contribute to coverage of the scratched area. In the case of UPCI-SCC090 cells, no cell migration was observed in control and BDNF-treated conditions, the cells died from the single 10-µg/mL MMC treatment (Appendix A). The UPCI-SCC090 cells were counted and their viability was investigated 96 h after the single MMC treatment in control and 25 ng/mL BDNF-treated conditions. The relation of the trypan-blue negative cells was over 90% in both control and 25 ng/mL BDNF-treated conditions, and it did not differ significantly (not shown). The UPCI-SCC090 cells 96 h after single MMC treatment both in control and in two-times 48 h 25 ng/mL BDNF treatment conditions were paraffin embedded and stained with rabbit monoclonal antibody against cleaved caspase-3. Both control and BDNF-treated conditions showed significant levels of cleaved caspase-3 positive cells (Figure 4). As seen on Figure 4 the BDNF-treated cells looked bigger both in nuclei and in whole cell size. The cell numbers of control and 25 ng/mL BDNF-treated cells 96 h after single MMC-treatment also did not differ significantly, but the cell numbers in BDNF-treated condition were slightly more (Figure 5A). Correspondingly, BDNF induced a slight increase in cell numbers of TrkB^+^ UPCI-SCC090 cells after a single MMC treatment, but it was not sufficient to rescue the cells from MMC-induced apoptosis. As displayed on Figure 4, BDNF induced a morphological change in UPCI-SCC090 cells and we were wondering if BDNF induces epithelial–mesenchymal transition (EMT). This function of BDNF could have been expected based on available references [16,18,22].

As mentioned before, after MMC-induced cell cycle arrest, UPCI-SCC090 cells were treated with 25 ng/mL BDNF for 2 × 2 days followed by embedding in agarose and in paraffin, sectioning, and were stained with mouse monoclonal anti-pan-cytokeratin antibody combined with rabbit monoclonal vimentin antibody. In both the control (Figure 6A–C) and BDNF-treated conditions (Figure 6D–F), all UPCI-SCC090 cells were cytokeratin^+^, and some scattered cells showed vimentin reaction in the control condition (Figure 6C, white arrows, G), while nearly all cells became vimentin^+^ after BDNF-treatment (Figure 6F, white arrows, H). These results revealed that BDNF could not counteract the MMC-induced apoptosis, but it induced EMT in UPCI-SCC090 cells.

### 2.5. Effects of BDNF Treatments in SCC-25 Cells

As described before, Detroit 562 cells did not show any single sign of TrkB^+^ reaction, but as visible on Figure 3D, in SCC-25 cell culture very few scattered cells were TrkB^+^. A more sensitive PCR reaction with 620 base pairs amplification product also amplified *NTRK2* in SCC-25 culture. It means, that there is an *NTRK2* gene expression potential in SCC-25 cells, which is in agreement with our previous data [18]. We were wondering if TrkB could have been induced/upregulated in SCC-25 HPV^-^ oral SCC cells. As evidenced in Figure 7, we found the way to induce TrkB in a limited cell population of SCC-25 cells. Two complementary methods, which use two different clones and production of TrkB-specific antibodies; cell signaling array analysis, and paraffin embedding and immunohistochemical reaction confirmed that TGF-β1 induced/upregulated a TrkB^+^ cell population in SCC-25 oral SCC cells. Moreover, as demonstrated in Figure 7A, TGF-β1 induced a package of growth factor receptors including epidermal growth factor receptor (EGFR), nerve growth factor receptor (TrkA), and fibroblast growth factor receptor (FGFR).

After recognizing that in SCC-25 cells a TrkB^+^ cell population was inducible by TGF-β1, we repeated the single MMC treatment as done in SCC-25 cells, but before MMC-treatment we gave 1 ng/mL TGF-β1 [20] or 25 ng/mL BDNF for 72 h to them. The single 30-min 10-µg/mL MMC treatment was followed by 2 times 48 h treatment with TGF-β1 or with BDNF. After the treatments, the cell numbers were counted and related to the original plated numbers. Treatment with TGF-β1 or BDNF did not significantly affect the cell numbers after MMC challenge (Figure 5B). TGF-β1 treatment before and after MMC-challenge resulted in a non-significant decrease in cell numbers. After all pre-treatment and post-treatment conditions, 1000 cells from all samples represented on panels A and B of Figure 5 were plated in 75 cm^2^-cell culture flasks and grown in serum-supplemented medium for three weeks and the growing clones were stained with gentian violet and were counted. In contrast to UPCI-SCC090 cells, where no clones were grown, the SCC-25 cells produced clones in all treatment conditions. The number of growing clones were related to plated cell numbers (1000). The number of growing clones was significantly lower in the case of TGF-β1 treatment than in all other settings (Figure 5C), but in combination of TGF-β1 treatment before MMC and BDNF after MMC the number of clones were significantly higher than in the case of TGF-β1 treatment both before and after MMC application. The UPCI-SCC090 cells did not form any clones three weeks after MMC treatment (Figure 5E) either in control or in BDNF-treated settings.

## 3. Discussion

Brain-derived neurotrophic factor was sufficiently available either in normal or in tumor tissue and also in cultured HNSCC cell lines. Interestingly, the specific high affinity BDNF receptor, TrkB, was limitedly available, only in 40% of HNSCC, 1 of 12 normal mucosa samples, in Detroit 562 HNSCC cell line without any sign of positive reaction, in SCC-25 oral SCC cells in extremely rare positive cells, but in UPCI-SCC090 HPV^+^ oropharynx SCC cells in all cells. In HNSCC tissue, the TrkB positive reaction was either focal or wide-spread diffuse within the tumor cell nest. Our data suggest that both in HNSCC cell lines and in HNSCC tissue, the high positive TrkB^+^ reaction, which was detected in 20% of all cases, is a result of an upregulation of normally arranged wild-type gene, and not a result of a rearrangement, which is comparable with the available references [23]. TrkB was more frequently positive in cases with more severe N-stage, in secondary tumors, recurrences, and in cases with altered p53. These results suggested, also in agreement with the available peer publications, that high TrkB-positivity might be indeed related with invasive tumor, and with therapy resistance [22] and with tumor cells with no apoptosis functionality (p53 altered).

For testing the cell migration and therapy resistance conditions the UPCI-SCC090 cells were first available, since they expressed the *NTRK2* gene and synthesized the protein product, TrkB. For analysis of cell migration the so-called scratch assay was used [20], which contains a Mitomycin C (MMC) treatment (a single 10-µg/mL treatment for 30 min at 37 °C) to ensure cell cycle arrest [40]. In this method, a pipette tip scratched gap is repopulated by cells, either by cell division or by cell migration. Mitomycin C is used to uncouple cell migration from cell division. In the case of UPCI-SCC090 cells, no cell migration was observed in control and BDNF-treated conditions; the cells died from the single 10-µg/mL MMC treatment. Again, the cells were also not rescued by BDNF, they died by apoptosis. The clonogenic survival assay further revealed that BDNF was not able to ensure the survival of any clones in the HPV^+^ cell line. In contrast, BDNF ensured morphological changes in UPCI-SCC090 cells, and induced EMT in the majority of the cells. These data suggest that in UPCI-SCC090 cells, BDNF treatment ensured EMT, which was not accompanied by chemotherapy resistance. These results are not contradictive, moreover, they confirm published evidence as Vishnoi et al. [41] published in 2016 that the HPV16 E6 protein actively participated in EMT, while Thomas et al. [42] published very recently the participation of HPV16 E7 protein in cisplatin sensitivity.

In SCC-25 cells a TGF-β1 pre-treatment ensured TrkB protein detection in some cell groups. The induction of TrkB was a part of the growth factor receptor upregulation package of TGF-β1-treatment, where also TrkA, EGFR, and FGFR were upregulated, an observation which is novel, but partly similar results have been published before [43].

The SCC-25 cells were active in migration and survived without a problem the single MMC challenge without BDNF-treatment as well, but BDNF was able to counteract the growth suppression effects of TGF-β1.

Our data based on observation of clinical and experimental approaches revealed that chemotherapeutic agents might be effective in cell death induction of HPV^+^ oropharynx SCC cells such as UPCI-SCC090. These cells synthesize the high affinity BDNF receptor TrkB, constitutive express *BDNF* and we added 25 ng/mL recombinant BDNF to these cells, all of these were not sufficient to counteract a single chemotherapeutic agent challenge in them. The first possible explanation for this phenomena could be that UPCI-SCC090 might have expressed the pro-apoptotic low affinity BDNF receptor, p75NTR, which might be activated by higher concentrations of BDNF [44,45]. In fact, this would be doubtful, as we have observed and published recently that UPCI-SCC090 cells do not contain any p75NTR^+^ cells [28]. At the same time, some cell growth advantages of BDNF have been evidenced in TGF-β1-treated SCC-25 cells, which also contain in scattered cells p75NTR [28]. It seems that the known therapy escape factor TGF-β1 [46] might render HPV^−^ HNSCC cells responsive to growth factors such as EGF, FGF, NGF or BDNF, which support expansion of cell clones that survive therapeutic interventions.

Taken together, HPV^+^ oropharynx squamous cell carcinoma, in agreement with available literature, has a preferred prognosis [26,41,42,47,48], partly based on effective apoptosis induction of therapeutic agents and inefficiency of survival factors such as BDNF. In HPV^−^ HNSCC the induction of apoptosis might be less efficient, as we also did not find apoptotic cells in MMC-treated SCC-25 culture, and a combination of survival factors as TGF-β1 and EGF, FGF, NGF or BDNF could support the survival of the treated cells. Multiple effects of different factors might contribute to improve cell survival of treated HPV^−^ HNSCC cells, and worse outcomes in the HPV^−^ HNSCC patients.

## 4. Materials and Methods

### 4.1. Patient Samples, Immunohistochemistry

The procedures followed were in accordance with the ethical standards of the committee on human experimentation of the institution and in accord with the Helsinki Declaration of 1975 as revised in 1983. Permission was obtained from the local ethics committee to collect pretreatment biopsy samples for molecular biological investigation, paraffin embedding, sectioning, and immunohistochemical analysis (Reference Number: UN4428 303/4.14, 26 July 2011). Informed consent was obtained from all patients. Immunohistochemical analysis of TrkB receptor was performed in 131 randomly selected specimens of incident locally advanced HNSCC treated between March 2010 and October 2017 at the Department of Otorhinolaryngology—Head and Neck Surgery, Medical University of Innsbruck. Clinical data are summarized in Table 2. As a control, 12 normal mucosa tissue samples derived from uvulopalatopharyngoplasties (UPPP) in patients with sleep apnea syndrome were included. Pretreatment tumor samples were obtained during diagnostic panendoscopy. Patient samples were paraffin embedded, sectioned, and immunostained as published recently [28].

### 4.2. In Situ Hybridization and Immunohistochemistry

In situ hybridization was performed on 5 µm paraffin sections in a Ventana Discovery Classic immunostainer (Tucson, AZ, USA) using Ribomap kit and Bluemap kit (Ventana) utilizing digoxigenin (DIG) labelled riboprobes as published previously [28]. The designed *BDNF* riboprobes have been published in detail before [49]. Immunohistochemistry was done on a Ventana Discovery staining automat using DABMap kit and rabbit monoclonal antibodies (EPR17341 as pan-Trk antibody from Abcam, Cambridge, UK; and 82G2 as specific anti-TrkB antibody from Cell Signaling Technology, Frankfurt am Main, Germany). Cleaved caspase-3 was detected using rabbit monoclonal primary antibody (Cat. Nr. 9664; Cell Signaling Technology). Immunofluorescence staining was performed using mouse monoclonal pan-cytokeratin antibody (cat. nr. 760-2595, Roche Ventana, Mannheim Germany) and anti-vimentin, clone SP20 rabbit polyclonal antibody (Abcam), Alexa Fluor^TM^ 488 or 594 conjugated secondary antibodies for detection of the immunoreaction were purchased from Molecular Probes (Life Technologies, Darmstadt, Germany). All antibodies were diluted as suggested by the manufacturers. The immunofluorescent-stained slides were cell nuclei counterstained by DAPI (Molecular Probes).

### 4.3. Image Analysis of Immunohistochemistry and In Situ Hybridization

The immunostained and riboprobes reacted sections were digitalized at 20× magnification utilizing a TissueFaxs Plus System coupled onto a Zeiss^®^ Axio Imager Z2 Microscope (Jena, Germany). Regions of interest were then acquired using the TissueFaxs (TissueGnostics^®^, Vienna, Austria). TrkB immunostaining was scored [50] (0: no staining, low (1): under 30% of cells positive, middle (2): 30–66% of cells positive, high (3): more than 66% of cells positive in cancer cell nests) and Mann–Whitney test was used to detect differences between HNSCC and UPPP. Dot Plots, frequency diagrams, and heat maps were created using TissueQuest (TissueGnostics^®^) [51,52].

### 4.4. Cell Lines

The SCC-25 and UPCI-SCC090 cells were acquired from the German Collection of Microorganisms and Cell Cultures (DSMZ, Braunschweig, Germany, DSMZ no.: ACC 617), SCC-25 cells were cultured in DMEM/F12 medium [53,54], UPCI-SCC090 [39] cells in EMEM medium supplemented with 10% FBS, 2 mM l-glutamine, 100 units/mL penicillin, and 100 μg/mL streptomycin [19]. Detroit 562 cells were purchased from Cell Lines Service (CLS, Eppelheim, Germany) and were cultured in EMEM medium [55] supplemented with 10% FBS, 2 mM l-glutamine, 100 units/mL penicillin, and 100 μg/mL streptomycin. For experimental purposes, cells were cultured in an albumin-containing medium where serum proteins were replaced by 4.4 g/L bovine serum albumin from PAA Laboratories, (Pasching, Austria).

### 4.5. Paraffin Embedding of Cultured Cells

Routinely cultured cell lines (2–4 × 10^6^) were collected by centrifugation and embedded as cell pellets in agarose as published before [28,56], and modified as follows: Cells were harvested by centrifugation at 290× *g* for 10 min at 4 °C, and the resulting pellet was fixed in 10 mL neutral buffered 4% formaldehyde solution (Flintsbach am Inn, Germany). After fixation the cells were centrifuged at 400× *g* for 10 min at room temperature. The cell pellet was resuspended in 300 µL PBS, transferred to Eppendorf tube (1.5 mL), and kept on ice. Low melting point agarose (with gelling temperature point 34–37 °C) was prepared in PBS as 3% solution in labor glassware by microwave warming and it was equilibrated in a thermoblock to 65 °C for at least 30 min. The 300 µL PBS-cell suspension was also equilibrated to 65 °C for not more than 10 min. Then, 600 µL melted equilibrated agarose was pipetted to the cell suspension, followed by spinning at 2000× *g* for 5 min at room temperature. After that, the tube was placed on ice, the cell pellet was trimmed, and it was placed in embedding cassette. The cell pellet in the cassette was stored in PBS containing 0.05–0.1% sodium azide until embedded in paraffin as published in detail before [28].

Similar to the tissue sections, from the cell pellets 5-µm thick sections have been cut. The cell sections did not contain any overlaps, and the cells were distributed. The cell sections were stained immunohistochemically exactly identical with the tissue sections. The percentage of positive cells for the required reaction was identified after scanning the sections in the TissueFaxs system and evaluating with Tissuequest software [28].

### 4.6. Cell Treatments

For the treatment with 0–50 ng/mL BDNF [13] or with 1 ng/mL TGF-β1 [20] (RnD Systems, Minneapolis, MN, USA), 6.7 × 10^4^ cells/mL were plated in serum containing medium [20] and cultured for 72 h. After that, the cells were washed with PBS and incubated with 10 µg/mL Mitomycin C (Sigma–Aldrich^®^, St. Louis, MI, USA) in serum-free albumin-containing medium for 30 min at 37 °C ensuring cell cycle arrest [40]. Then the cells were washed twice with albumin-containing medium and subsequently treated with albumin-containing medium for two times 48 h supplied with 25 ng/mL recombinant human BDNF [13] for altogether 96 h. After completion of treatments, the cells were used for cell counting with trypan blue staining (Sigma, Darmstadt, Germany) in a Neubauer chamber (Paul Marienfeld GmbH & Co. KG, Lauda-Königshofen, Germany). One-thousand cells from all samples were plated in 75-cm^2^ cell culture flasks grown in serum-supplemented medium for 3 weeks and the growing colonies were stained with gentian violet and were counted. The number of growing colonies were related to plated cell numbers (1000) [57].

During the treatments cells were filmed by a Juli BR live cell imaging system (Peqlab, Erlangen, Germany), which was also used for estimation of the percentage of covered area in the culture dish.

### 4.7. RNA Isolation, PCR, and Sequencing

For RNA isolation, 2–4 × 10^6^ cells or 2–3-mm tissue slices were collected and lysed in 1 mL TRIzol^®^ Reagent (Ambion^®^, Life technologies^TM^, Carlsbad, CA, USA), and RNA was isolated as instructed by the manufacturer of TRIzol. The RNA concentrations were determined by photometric measurements (BioPhotometer plus 6132, Eppendorf, Germany). Total RNA was reverse transcribed by M-MuLV Reverse Transcriptase (GeneON, Ludwigshafen am Rhein, Germany) in a MyiQ^TM^ cycler (BIO-RAD Laboratories, Inc., Hercules, CA, USA) following the manufacturer’s instructions. PCR of cDNA transcripts was performed in a MyiQ^TM^ cycler (BIO-RAD Laboratories, Inc.) using Go–Taq master mix (Promega, Madison, WI, USA) and the following forward: 5′-GGC TGA CAC TTT CGA ACA CA-3′ and reverse: 5′-CTT ATG AAT CGC CAG CCA AT-3′ primers for *BDNF*, and forward: 5′-CTA GGG ATG TCG TCC TGG ATA-3′; reverse: 5′-AGG GCC CTA GCC TAG AAT GTC-3′ for *NTRK2* 2528 base pairs, forward: 5′-ATC TCC AAC CTC AGA CCA CC-3′; reverse: 5′-CTT ACA TGG CAG CAT CAA CCA-3′ for *NTRK2* 620 base pairs, and forward: 5′-CCG AAA GTT GCC TTT TAT GGC T-3′; reverse: 5´-AGG TCT CAA ACA TGA TCT GGG T-3′ for the housekeeping gene *ACTB*. The primers were synthesized by Invitrogen^TM^ (Darmstadt, Germany). The PCR setup and the cycling conditions were instructed in the manual of Go-Taq. The PCR products were electrophoresed in 1% agarose run in Tris–Acetate–EDTA buffer for one hour at 100 volts. Gels were photographed in an Azure C500 (Azure Biosystems, Dublin, CA, USA). The PCR products were cleaned from the agarose gel by Minelute Gel elution kit (Qiagen, Stockach, Germany) following the instructions of the manufacturer, and were sent for Sanger sequencing to Microsynth (Balgach, Switzerland).

### 4.8. Protein Analysis in Cell Signaling Array

Two × 10^5^ SCC-25 cells/10 cm dish were plated for three days. Then an adaption from serum to albumin medium followed over two days. After that the cells were treated with albumin medium supplemented with TGF-β1 (1 ng/mL) for 24 h. The next day the cells were used for protein extraction in Cell Lysis Buffer (Cell Signaling Technology). The concentration of the samples was measured, and they were diluted to the same concentration of 1 µg/µL. After that the cell lysates were processed on a PathScan^®^ RTK Signaling Antibody Array Kit (Cell Signaling Technology^®^, #7949) according to the manufacturer’s instructions.

### 4.9. Statistical Analysis

Quantitative data were analyzed for their distribution by D’Agostino and Pearson normality test. In case of normal distribution, two data sets were compared by unpaired *t*-test, more data sets were compared by Dunnett’s multiple comparisons test. If data were not normally distributed, two data sets were compared by Mann–Whitney U test, and more data sets were compared by Tukey’s or Dunn’s multiple comparisons test. Data of different categories were represented as mean ± standard error of measurement in column bars or in dot plots. Statistical significance was claimed by *p* < 0.05. Patient survival of HPV^+^ and HPV^−^ cases were compared by Kaplan–Mayer survival analysis and log-rank (Mantel–Cox), Breslow and Tarone–Ware pairwise comparisons. Statistical analysis and graph presentations were done in Graphpad Prism 7.00 and in IBM SPSS vers. 24.

## Figures and Tables

**Figure 1 ijms-20-00272-f001:**
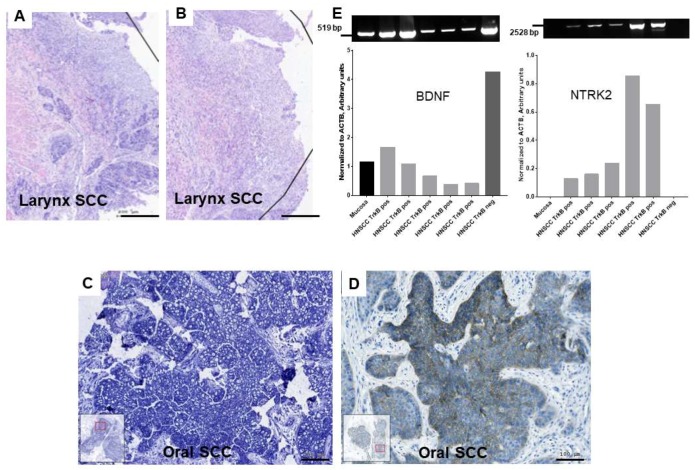
mRNA expression of brain-derived neurotrophic factor (BDNF) and NTRK2, protein synthesis of NTRK2 in head and neck squamous cell carcinoma (HNSCC) (**A**,**B**): In situ hybridization of antisense (**A**) and sense (**B**) riboprobe for *BDNF* (blue) in larynx SCC, cell nuclei counterstained in nuclear fast red. The antisense probe shows intensive purple—blue reactive areas, while the tissue reacted with the sense probe is slightly purple—blue stained. (**C**): In situ hybridization of antisense *BDNF* riboprobe and (**D**): immunohistochemistry of TrkB (brown) in tumor cell nests of oral SCC. **A** and **B** and **C** and **D** are sequential sections. (**E**): PCR detection of *BDNF* (519 base pairs, bps), *NTRK2* (full protein coding area, 2528 bps) normalized to loading control *ACTB* (534 bps, not shown, normalized values represented as column bars) gene expression in cDNA samples of control UPPP normal mucosa, immunohistochemically (IHC) TrkB-positive and TrkB-negative HNSCC. *BDNF* is expressed in both normal and malignant tissue, *NTRK2* is not present in normal mucosa, but if positive TrkB IHC staining was detected, the *NTRK2* gene expression was also confirmed by PCR, while TrkB-negative IHC was also negative in RT-PCR. (**A**–**D**) images were taken by the TissueFaxs system, bars: 200 µm: (**A**,**B**); 100 µm: (**C**,**D**). Bands densitometry was done using Azurespot 14.2.

**Figure 2 ijms-20-00272-f002:**
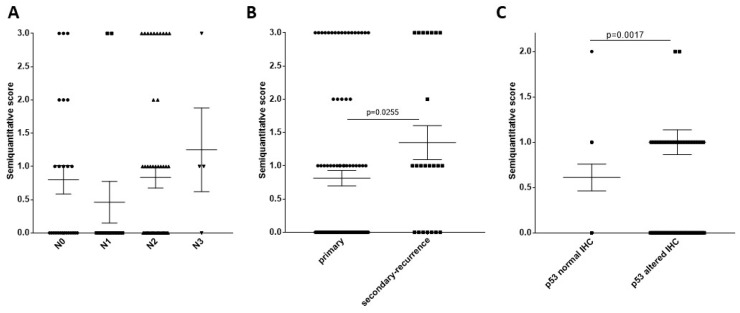
Relationship of TrkB immunostaining with N-stage, recurrence or second tumor and with p53 immunohistochemistry. TrkB was detected in 56 of 131 HNSCC cases and was scored semi-quantitatively as described in the Materials and Methods section. The semiquantitative values did not show normal distribution as tested by D’Agostino and Pearson normality test. The IHC score values were displayed as dot plots of mean and standard error of measurement (SEM). (**A**) N-stage data was available in 96 cases with the following case numbers: N0: 25, N1: 13, N2: 54 and N3: 4. N1–N3 showing increasing severity of lymph node metastasis also showed increase tendency in the mean value of the frequency of TrkB^+^ cases, but this difference was not statistically significant using Dunn’s multiple comparisons test. (**B**) Comparing primary (*n* = 97) and secondary or recurrent HNSCC (*n* = 23) the TrkB staining was significantly more frequent in secondary or recurrent HNSCC than in primary tumor. Statistical comparison was performed using Mann–Whitney test. (**C**) Comparing HNSCC with normal (wild-type; *n* = 54) TP53 and with altered (*n* = 76) TP53 immunohistochemistry, the TrkB staining was significantly more frequent in cases with altered TP53 than in cases with normal TP53. Statistical comparison was performed using Mann–Whitney test.

**Figure 3 ijms-20-00272-f003:**
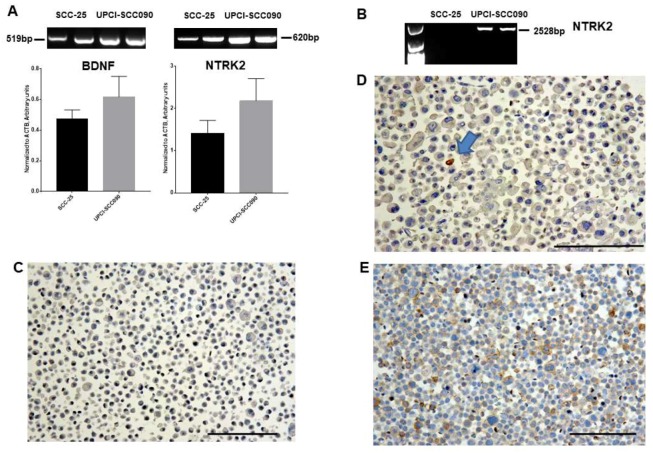
Gene expression of *BDNF* and *NTRK2* in SCC-25 and UPCI-SCC090 cells, and immunohistochemical reaction of TrkB in paraffin embedded HNSCC cell lines. (**A**) Using an optimized primer designed for a 519 bp PCR product of *BDNF* and for a 620 bp PCR product of *NTRK2* gene expression, both were detected in both SCC-25 and UPCI-SCC090 HNSCC cell lines. *BDNF* and *NTRK2* levels were higher in UPCI-SCC090 cells. The column bars represent mean ± SEM. Statistical comparison is not available due to the relative low number of measurements. *ACTB* was used as loading control and its optical density values were used for normalization of both *BDNF* and *NTRK2* optical densities. (**B**) If whole protein coding region of corresponding 2528 base pairs of *NTRK2* was amplified, only in UPCI-SCC090 cells was a PCR product available. At protein level using paraffin embedded cells, TrkB was not present in Detroit 562 (**C**), and was only in few scattered cells in SCC-25 ((**D**), arrow) cell line, but was present in all cells in UPCI-SCC090 (**E**) cell line. TrkB was stained using the rabbit polyclonal 80G2 antibody and was detected in brown. Cell nuclei were counterstained in blue by hematoxylin. Bars: 100 µm.

**Figure 4 ijms-20-00272-f004:**
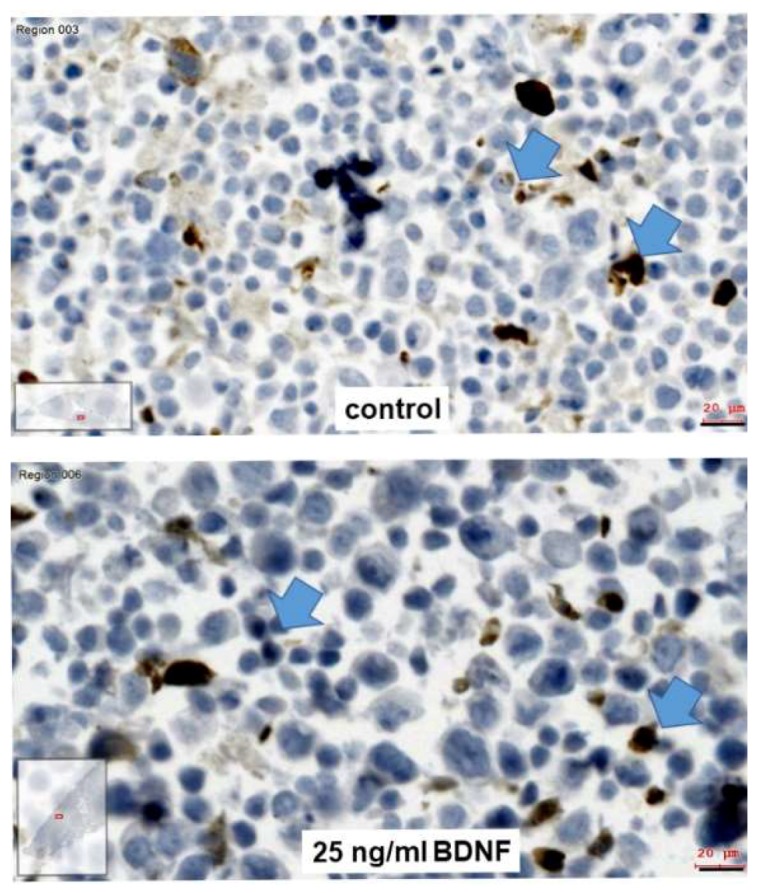
Apoptosis detection in paraffin-embedded UPCI-SCC090 cells by immunohistochemical reaction of cleaved caspase-3 in control and 25 ng/mL BDNF-treated conditions UPCI-SCC090 cells were counted, paraffin embedded, sectioned, and stained with cleaved caspase-3 specific rabbit monoclonal antibody 96 h after a single MMC treatment in control and 25 ng/mL BDNF-treated conditions. The antibody reaction was detected in brown, cell nuclei were counterstained with hematoxylin in blue. Both control and BDNF-treated cells show a significant portion of dark brown cleaved-caspase-3-positive apoptotic cells (blue arrows). Bars: 20 µm. Images were taken with the TissueFaxs ^®^ system. The white boxes in the left corner of the images represent the whole section, the red part in these white boxes shows the position of the imaged area within the section.

**Figure 5 ijms-20-00272-f005:**
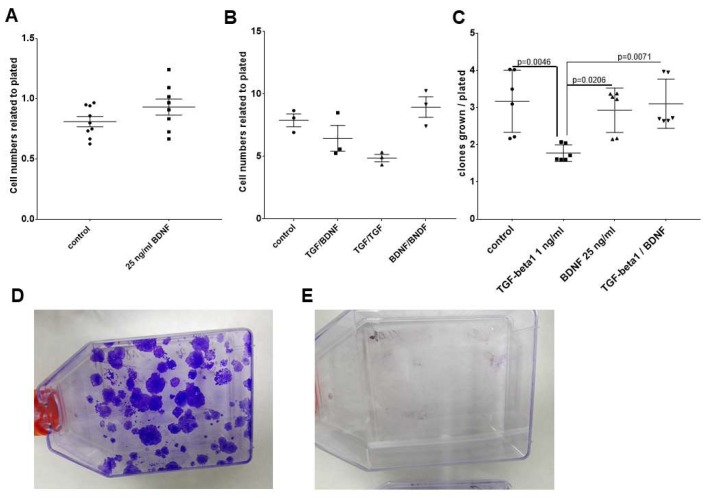
Influence of BDNF on cell survival of UPCI-SCC090 and SCC-25 cells after MMC treatment UPCI-SCC090 and SCC-25 cells were plated in 6-well plates in serum-free medium or in medium supplemented with 1 ng/mL TGF-β1 (for SCC-25) or 25 ng/mL BDNF (for both) and cultured for 72 h followed by 30 min 10 µg/mL MMC treatment and 2-times 48 h treatment with TGF-β1 (for SCC-25) or BDNF. After the treatments, the cell numbers were counted and related to the original plated numbers ((**A**) UPCI-SCC090, (**B**) SCC25). One-thousand cells from each sample, represented on panels (**A**,**B**), were plated in 75-cm^2^ cell culture flasks grown in serum-supplemented medium for 3 weeks and the growing colonies were stained with gentian violet and were counted. The number of growing colonies were related to plated cell numbers (1000). Treatment of BDNF or TGF-β1 did not cause any significant difference in the MMC surviving cell numbers of UPCI-SCC090 cells (**A**) or SCC-25 (**B**), BDNF treatment showed a non-significant increase in the cell numbers of both cell lines. The cell numbers data were normal distributed in UPCI-SCC090 cells, unpaired *t*-test did not show significant difference for BDNF-treatment. In SCC-25 cells the cell numbers data were not normally distributed, and Dunn’s multiple comparisons test did not show any significant difference for the treatments against the control. In case of SCC-25 cells, after three weeks, cell clones were growing in control and treated settings (**C**,**D**), the number of growing clones (stained in blue by gentian violet on **D**) was significantly lower in case of TGF-β1 treatment than in all other settings (**C**), but in combination of TGF-β1 treatment before MMC and BDNF after MMC the number of clones were significantly higher than in case of TGF-β1 treatment both before and after MMC application. UPCI-SCC090 cells did not form any clones three weeks after MMC treatment (**E**) either in control or in BDNF-treated settings. The number of growing clones related to plated cell numbers did not show normal distribution. Significance was claimed if the *p*-value was lower than 0.05 using Tukey’s multiple comparisons test.

**Figure 6 ijms-20-00272-f006:**
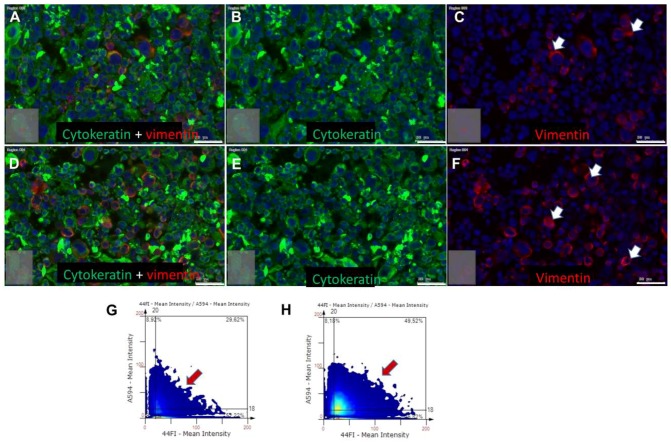
BDNF-treatment induces EMT in UPCI-SCC090 cells. UPCI-SCC090 cells were plated in serum-containing medium, treated with 10 µg/mL MMC for 30 min followed by 25 ng/mL BDNF-treatment for 2 × 2 days. After completion of this procedure, cells were collected, embedded in agarose and in paraffin, 5 µm sections were cut, and these were stained with mouse monoclonal anti-pan-cytokeratin antibody (Ventana, detected in green) combined with rabbit monoclonal vimentin (SP-20) antibody (detected in red), and 4′,6-Diamidin-2-phenylindol (DAPI) (blue) cell nuclear counterstaining. The immunohistochemical reactions are shown combined (**A**,**D**) and separated (**B**,**C**,**E**,**F**) in control (**A**–**C**) and BDNF-treated (**D**–**F**) cells. The camera profile, fluorescence excitation light exposition conditions were the same by all images, and experiment profile was re-used. The BDNF-treatment lead to increased vimentin representation without the loss of cytokeratin signal. The mean intensity for the cytokeratin (detected by the 44Fl Zeiss Filter Channel) and vimentin (detected by the A594 Zeiss Filter Channel) signals in all cells have been plotted on color-mapped heat intensity diagrams in control (**G**) and in BDNF-treated conditions (**H**). These graphs also show an increase in the double-positive (cytokeratin–vimentin) cell population (**G**,**H:** red arrows; **C**,**F**: white arrows). Bars: 50 µm.

**Figure 7 ijms-20-00272-f007:**
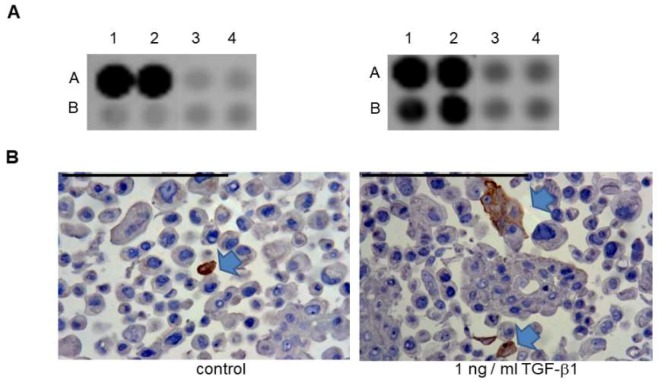
TGF-β1 regulates the protein levels of TrkB, together with epidermal growth factor receptor (EGFR) and TrkA in SCC-25 cells. (**A**) 2 × 10^5^ SCC-25 cells/10-cm dish were plated for three days. Then an adaption from serum to albumin medium followed over two days. After that the cells were treated with albumin medium supplemented with TGF-β1 (1 ng/mL) for 24 h. The next day the cells were used for protein extraction in Cell Lysis Buffer (Cell Signaling Technology). The concentration of the samples was measured, and they were diluted to the same concentration of 1 µg/µL. After that the cell lysates were processed on a PathScan^®^ RTK Signaling Antibody Array Kit (Cell Signaling Technology^®^, #7949) according to the manufacturer’s instructions. A1–A2: positive control; B1–B2: EGFR (Her1); A3–A4: TrkA; B3–B4: TrkB. (**B**) 5 × 10^4^ SCC-25 cells/mL were plated in 75-cm^2^ culture dishes (Orange Scientific, Braine-l’Alleud, Belgium,) in serum-free, albumin containing medium [20] and cultured for 72 h. After that, the cells were washed with PBS and incubated with 10 µg/mL Mitomycin C (Sigma–Aldrich^®^, St. Louis, MI, USA) in serum-free albumin-containing medium for 30 min at 37 °C ensuring cell cycle arrest. Then the cells were washed twice with albumin-containing medium and subsequently treated with albumin-containing medium for two times 48 h supplied with 1 ng/mL recombinant human TGF-β1, or in control conditions without TGF-β1 for 96 h. After completion of treatments the cells were embedded in agarose and in paraffin as described before [28]. TrkB was stained using the rabbit polyclonal 80G2 antibody and was detected in brown. Cell nuclei were counterstained in blue by hematoxylin. Arrows represent positive cells. Bars: 100 µm.

**Table 1 ijms-20-00272-t001:** Descriptive statistics (frequency distribution) of the neurotrophin receptor tyrosine kinase-B (TrkB) staining in normal mucosa and head and neck squamous cell carcinoma (HNSCC) samples.

Tissue Type	Negative	Focal Staining Pattern	Diffuse Staining Pattern	Total
HNSCC	75	30	26	131
Normal tissue	11	1	0	12
Total	87	31	26	144

**Table 2 ijms-20-00272-t002:** Patients data. Mean age of patients: 62.33 ± 11.3 years; minimum: 31; maximum: 91. In oropharynx samples 44 cases (61.1%) were HPV^−^ and 28 cases (38.9%) were HPV^+^.

Tissue Type	Number of Cases	Percent/%
HNSCC	131	91.0
UPPP epithelium	12	9.0
**Tumor Localization**	**Number of Cases**	**Percent/%**
oral	15	11.5
nasopharynx	3	2.3
oropharynx	72	55
larynx	23	17.6
hypopharynx	15	11.5
other	3	2.2
Total	131	100.0
**Gender**	**Frequency**	**Percent/%**
Male	102	77.9
Female	29	22.1
Total	105	100.0
**HPV**	**Frequency**	**Percent/%**
Positive	33	25.2
Negative	98	74.8
Total	131	100.0

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
