# Peer review of "Brain-Derived Neurotrophin and TrkB in Head and Neck Squamous Cell Carcinoma"

_ijms, 2019, doi:10.3390/ijms20020272_

Round 1

Reviewer 1 Report

In this work, the authors attempt to evaluate whether BDNF and TrkB play roles in tumor cell survival, invasion, and therapy resistance in HNSCC. Using a series of immunohistochemistry, in situ hybridization and qPCR studies in patient samples and in vitro experiments in cell lines exposed to TGFb or MMC, they conclude that BDNF is likely ineffective HPV+ HNSCC patients but BDNF and TGFb improve tumor cell survival and contribute to worse patient prognosis.

Overall, the manuscript is well written and the conclusions are –for the most part- are supported from the findings. A major criticism is that the experiments addressing the functional relevance of BDNF and TrkB in tumor initiation in HNSCC are rather limited. But the work is appreciated for its clinical relevance and the attempt to complement some of the findings in patients samples with cell lines.

Some points could improve the readability of the manuscript:

-Please indicate the normal and tumor samples on the upper right/left corner of the respective images (besides the figure legend).

-Arrowheads on the images could indicate the areas of interest.

-The qPCR images would benefit from a more quantitative presentation using column bars with +/- st.dev. (next to the respective images).

 -There is no information on the statistical analysis used nor whether the authors use a standard deviation or standard error of means in respective graphs using column bars.

-More information on SCC-25, Detroit 562 and UPCI1-SCC090 cells should be provided (what these cell lines are, what the acronyms are, why they were chosen, and how they relate to each other and to this work.

Author Response

„In this work, the authors attempt to evaluate whether BDNF and TrkB play roles in tumor cell survival, invasion, and therapy resistance in HNSCC. Using a series of immunohistochemistry, in situ hybridization and qPCR studies in patient samples and in vitro experiments in cell lines exposed to TGFb or MMC, they conclude that BDNF is likely ineffective HPV+ HNSCC patients but BDNF and TGFb improve tumor cell survival and contribute to worse patient prognosis.”

„Overall, the manuscript is well written and the conclusions are –for the most part- are supported from the findings. A major criticism is that the experiments addressing the functional relevance of BDNF and TrkB in tumor initiation in HNSCC are rather limited. But the work is appreciated for its clinical relevance and the attempt to complement some of the findings in patients samples with cell lines.”

„ Some points could improve the readability of the manuscript”:

1.       „Please indicate the normal and tumor samples on the upper right/left corner of the respective images (besides the figure legend).”

Author response: This suggestion has been fulfilled by indication of the normal and tumor sample origin on Figure 1.

2.       „Arrowheads on the images could indicate the areas of interest.”

Author response: This suggestion has been fulfilled by including arrows on Figures 3, 4, 6, 7. 

3.       „The qPCR images would benefit from a more quantitative presentation using column bars with +/- st.dev. (next to the respective images).”

Author response: The used PCR reaction was not a quantitative PCR, it was an endpoint PCR reaction combined with gel electrophoresis in a qualitative view to indicate a detectable gene expression. Nevetheless, following the suggestion of the Reviewer, the suggested column bars of the normalized gene expression of  BDNF and NTRK2 genes representig mean +/- standerd error of measurement have been included in the revised manuscript (Figures 1, 3).

4.       „There is no information on the statistical analysis used nor whether the authors use a standard deviation or standard error of means in respective graphs using column bars.”

Author response: Authors are grateful for the constructive suggestion, the required details are added now into the Figure captions of Figure 2, 3 and 5. If the suggested column bars were used they represent mean +/- standerd error of measurement. Following the biostatistical review of this manuscript by Reviewer 2, Figures 2 and 5 contain dotplots.

5.       „More information on SCC-25, Detroit 562 and UPCI1-SCC090 cells should be provided (what these cell lines are, what the acronyms are, why they were chosen, and how they relate to each other and to this work.”

Author response: Authors are grateful for the constructive suggestion, the required details are added now in lines 212-223, and as follows:

SCC-25 cells were originally isolated from the primary tumor of a patient with tongue carcinoma [18, 31]. SCC-25 cells are from primary oral SCC, maintained in in vitro cultures, might be also xenografted, but they grow only in severe combined immunodeficiency (SCID) mice and not in athymic mice. SCC-25 cells did not show metastatic potential in mouse xenograft models [32]. In vitro, SCC-25 cells were published to be radioresistant [33]. Detroit 562 cells are metastatic HPV-negative HNSCC cells, which demonstrated radio and chemoresistance [34, 35], and it is also a potential metastatic cell line in xenograft models [36]. Detroit 562 cells were originally isolated from the malignant pleural effusion of an OSCC [37, 38]. The UPCI-SCC090 cell line has been established by Robert Ferris and co-workers [39]. UPCI-SCC090 cells contain genome integrated HPV-16 DNA. The UPCI-SCC090 cells synthesize both p16INK4 surrogate HPV marker protein and E6, E7 HPV oncogene product proteins [39]. [39]. These cell lines enabled the investigation of primary and metastatic as well as HPV-positive and negative HNSCC models.

Reviewer 2 Report

This is a biostatistical review.

1. The authors need to add a statistical method section into the manuscript. In particular, (i) please provide the statistical methods used to test in Figures 2, 5, and Suppl Fig 2; (ii) please indicate whether the normality assumption was checked and, if violated, what remedy was used (such as, transformation); (iii) please state whether the multiple comparison correction was performed or not.

2. Suppl Fig 2. Please provide the numbers of risk as well as the group information. In addition, the current legend is not clear for me. I think the bottom two legends for the censored data) are redundant.

3. Figs 2 and 5: Please use either the boxplot or the dotplot.

3. Fig 2A. Please perform the multiple comparison correction. It appears that the difference between N1 and N3 might not be significant after the multiple comparison correction.

4. Fig 5. Please provide the more detailed legend. It should be stand-alone without looking at the main text.

5. LN 260-269: It seems that the figure number is missing (i.e., Figure 5).

Author Response

1.       „The authors need to add a statistical method section into the manuscript. In particular, (i) please provide the statistical methods used to test in Figures 2, 5, and Suppl Fig 2; (ii) please indicate whether the normality assumption was checked and, if violated, what remedy was used (such as, transformation); (iii) please state whether the multiple comparison correction was performed or not.”

Author response: The required section has been included (Page 16, lines 556-564) and also the required corresponding statistical infomation is now given to the Figure Captions. By more than 2 datasets multiple comparisons have been performed.  

2.       „Suppl Fig 2. Please provide the numbers of risk as well as the group information. In addition, the current legend is not clear for me. I think the bottom two legends for the censored data) are redundant.”

Author response: The required changes have been performed on Suppl. Fig. 2. 

3.       „Figs 2 and 5: Please use either the boxplot or the dotplot.”

In the mentioned figures the dotplot with S. E. of measurement has been used.

4.       „Fig 2A. Please perform the multiple comparison correction. It appears that the difference between N1 and N3 might not be significant after the multiple comparison correction.”

Author response: As suggested by the Reviewer, the Dunn's multiple comparisons test has been used, and the difference between N1 and N3 was not significant by this test.

5.       „Fig 5. Please provide the more detailed legend. It should be stand-alone without looking at the main text.”

Author response: the figure legend has been improved it is stand alone now.

6.       „LN 260-269: It seems that the figure number is missing (i.e., Figure 5).”

Author response: This area was the figure legend, which was not formatted correctly and confused the reviewer. This error has been corrected.